# Beyond Categorical Label Representations for Image Classification

**Boyuan Chen, Yu Li, Sunand Raghupathi, Hod Lipson**
Columbia University
`https://www.creativemachineslab.com/label-representation.html`

## Abstract

We find that the way we choose to represent data labels can have a profound effect on the quality of trained models. For example, training an image classifier to regress audio labels rather than traditional categorical probabilities produces a more reliable classification. This result is surprising, considering that audio labels are more complex than simpler numerical probabilities or text. We hypothesize that high dimensional, high entropy label representations are generally more useful because they provide a stronger error signal. We support this hypothesis with evidence from various label representations including constant matrices, spectrograms, shuffled spectrograms, Gaussian mixtures, and uniform random matrices of various dimensionalities. Our experiments reveal that high dimensional, high entropy labels achieve comparable accuracy to text (categorical) labels on the standard image classification task, but features learned through our label representations exhibit more robustness under various adversarial attacks and better effectiveness with a limited amount of training data. These results suggest that label representation may play a more important role than previously thought.

## 1 Introduction

Image classification is a well-established task in machine learning. The standard approach takes an input image and predicts a categorical distribution over the given classes. The most popular method to train these neural network is through a cross-entropy loss with backpropagation. Deep convolutional neural networks (Lecun et al., 1998; Krizhevsky et al., 2012; Simonyan & Zisserman, 2014; He et al., 2015; Huang et al., 2016) have achieved extraordinary performance on this task, while some even surpass human level performance. However, is this a solved problem? The state-of-the-art performance commonly relies on large amounts of training data (Krizhevsky, 2009; Russakovsky et al., 2015; Kuznetsova et al., 2018), and there exist many examples of networks with good performance that fail on images with imperceptible adversarial perturbations (Biggio et al., 2013; Szegedy et al., 2013; Nguyen et al., 2014).

Much progress has been made in domains such as few-shot learning and meta-learning to improve the data efficiency of neural networks. There is also a large body of research addressing the challenge of adversarial defense. Most efforts have focused on improving optimization methods, weight initialization, architecture design, and data preprocessing. In this work, we find that simply replacing standard categorical labels with high dimensional, high entropy variants (e.g. an audio spectrogram pronouncing the name of the class) can lead to interesting properties such as improved robustness and efficiency, without a loss of accuracy.

Our research is inspired by key observations from human learning. Humans appear to learn to recognize new objects from few examples, and are not easily fooled by the types of adversarial perturbations applied to current neural networks. There could be many reasons for the discrepancy between how humans and machines learn. One significant aspect is that humans do not output categorical probabilities on all known categories. A child shown a picture of a dog and asked "what is this a picture of?" will directly speak out the answer — "dog." Similarly, a child being trained by a parent is shown a picture and then provided the associated label in the form of speech. These observations raise the question: Are we supervising neural networks on the best modality?

Figure 1: Label Representations beyond Categorical Probabilities: We study the role of label representation in training neural networks for image classification. We find that high-dimensional labels with high entropy lead to more robust and data-efficient feature learning.

In this paper, we take one step closer to understanding the role of label representations inside the training pipeline of deep neural networks. However, while useful properties emerge by utilizing various label representations, we do not attempt to achieve state-of-the-art performance over these metrics. Rather, we hope to provide a novel research perspective on the standard setup. Therefore, our study is not mutually exclusive with previous research on improving adversarial robustness and data efficiency.

An overview of our approach is shown in Figure 1. We first follow the above natural observation and modify the existing image classifiers to "speak out" the predictions instead of outputting a categorical distribution. Our initial experiments show surprising results: that neural networks trained with speech labels learn more robust features against adversarial attacks, and are more data-efficient when only less than $20\%$ of training data is available.

Furthermore, we hypothesize that the improvements from the speech label representation come from its property as a specific type of high-dimensional object. To test our hypothesis, we performed a large-scale systematic study with various other high-dimensional label representations including constant matrices, speech spectrograms, shuffled speech spectrograms, composition of Gaussians, and high dimensional and low dimensional uniform random vectors. Our experimental results show that high-dimensional label representations with high entropy generally lead to robust and data efficient network training. We believe that our findings suggest a significant role of label representations which has been largely unexplored when considering the training of deep neural networks.

Our contributions are three fold. First, we introduce a new paradigm for the image classification task by using speech as the supervised signal. We demonstrate that speech models can achieve comparable performance to traditional models that rely on categorical outputs. Second, we quantitatively show that high-dimensional label representations with high entropy (e.g. audio spectrograms and composition of Gaussians) produce more robust and data-efficient neural networks, while high-dimensional labels with low entropy (e.g. constant matrices) and low-dimensional labels with high entropy do not have these benefits and may even lead to worse performance. Finally, we present a set of quantitative and qualitative analyses to systematically study and understand the learned feature representations of our networks. Our visualizations suggest that speech labels encourage learning more discriminative features.

## 2 RELATED WORKS

**Data Efficiency and Robustness** Data efficiency has been a widely studied problem within the context of few-shot learning and meta-learning (Thrun & Pratt, 2012; Vilalta & Drissi, 2002; Vanschoren, 2018; Wang et al., 2019). Researchers have made exciting progress on improving methods of optimization (Ravi & Larochelle, 2016; Li et al., 2017), weight initialization (Finn et al., 2017; Ravi & Larochelle, 2017), and architecture design (Santoro et al., 2016a;b).

There is also a large body of research addressing the challenge of adversarial defense. Adversarial training is perhaps the most common measure against adversarial attacks (Goodfellow et al., 2014; Kurakin et al., 2016; Szegedy et al., 2013; Shaham et al., 2018; Madry et al., 2017). Recent works try to tackle the problem by leveraging GANs (Samangouei et al., 2018), detecting adversarial examples (Meng & Chen, 2017; Lu et al., 2017; Metzen et al., 2017), and denoising or reconstruction (Song et al., 2017; Liao et al., 2017).

Most of these techniques for improving data efficiency and adversarial robustness study the problem from the perspective of model, optimizer and data. Relatively little research has been conducted on the labels themselves or more specifically, their representation. Hosseini et al. (2017) augmented categorical labels with a new NULL class to allow the model to classify and reject perturbed examples. Papernot et al. (2015) utilizes model distillation (Hinton et al., 2015) for adversarial defense. Papernot & McDaniel (2017) further augment the label used to train the distilled model with the predictive uncertainty from the original model. Nevertheless, the method of Hosseini et al. (2017) requires adversarial examples to train on while that of defensive distillation has been shown to be vulnerable to substitute model black-box attacks (Papernot et al., 2017).

**Label Smoothing** The closest approach to our work is Label Smoothing (LS) (Szegedy et al., 2016). Here we highlight key differences between our approach and LS. First, LS applies to discriminative outputs where both correct and incorrect class information are presented during training, while our output is generative and only correct class information is presented. That is, our outputs are not a distribution over classes. Although LS has been shown to improve adversarial robustness (Goibert & Dohmatob, 2020), it has not been shown to be effective for low-data learning. As we will show in our experiments, LS does not help when the amount of training data is limited, while our label representations lead to significant improvements. Therefore, our high-dimensional, high-entropy labels provide benefits beyond those provided by label smoothing.

## 3 BEYOND ACCURACY: EMERGENCE OF ROBUSTNESS AND EFFICIENCY

It is well-known that deep neural networks with similar accuracy on the same task may perform very differently under different evaluation scenarios. Additionally, real-world applications rely on more considerations than just accuracy. Robustness and data efficiency are two practical challenges for deep neural networks. We test the emergence of those properties under various label representations.

### 3.1 ROBUSTNESS UNDER ADVERSARIAL ATTACKS

We evaluate the robustness of all the trained models using the fast gradient sign method (FGSM) (Goodfellow et al., 2014) and the iterative method (Kurakin et al., 2016) across multiple widely used convolutional networks. We choose these attacks because their adversarial images $I_{\text{adv}}$ are usually indistinguishable from the original images $I$ or do not significantly affect human evaluation, but they can be very challenging for neural networks to correctly classify. When a loss function $J$ is involved in generating the adversarial images, we use the cross-entropy loss for the text model and the smooth L1 loss for the speech model.

**FGSM** is a fast one-step attack that generates an adversarial image by adding a small adversarial perturbation to the original image. The perturbation is based on the gradient of the loss with respect to the original image. The maximum magnitude of the perturbation is maintained by $\epsilon$:

$$\|I - I_{\text{adv}}\|_\infty \le \epsilon. \tag{1}$$

We test both untargeted and targeted versions of FGSM. The untargeted attacks increase the loss between the predicted class and the true class $Y_{\text{true}}$:

$$I_{\text{adv}} = I + \epsilon \cdot \text{Sign}(\nabla_I J(I, Y_{\text{true}})), \tag{2}$$

whereas the targeted attacks decrease the loss between the predicted class and a target class $Y_{\text{target}}$:

$$I_{\text{adv}} = I - \epsilon \cdot \text{Sign}(\nabla_I J(I, Y_{\text{target}})). \tag{3}$$

We choose a random incorrect class as the target class for each input image, and the same target classes are used to test different models. All $I_{\text{adv}}$ are normalized after the perturbation.

**Iterative Method** As an extension to FGSM, the iterative method applies multiple steps of gradient-based updates. In our experiments, we initialize the adversarial image $I_{\text{adv}}$ to be the original image $I$ so that $I_{\text{adv}}^0 = I$. Then we apply FGSM for 5 times with a small step size $\alpha = \epsilon/5$. The untargeted update for each iteration becomes

$$I_{\text{adv}}^{N+1} = \text{Clip}_{I,\epsilon}\left\{I_{\text{adv}}^N + \alpha \cdot \text{Sign}(\nabla_I J(I_{\text{adv}}^N, Y_{\text{true}}))\right\}, \tag{4}$$

and the targeted update becomes

$$I_{\text{adv}}^{N+1} = \text{Clip}_{I,\epsilon}\left\{I_{\text{adv}}^N - \alpha \cdot \text{Sign}(\nabla_I J(I_{\text{adv}}^N, Y_{\text{target}}))\right\}, \tag{5}$$

where $\text{Clip}_{I,\epsilon}$ denotes clipping the total perturbation $I_{\text{adv}}^N - I$ in the range of $[-\epsilon, \epsilon]$. We use the same targeted classes from FGSM for the evaluation on the iterative method.

## 3.2 Learning Efficiency with Limited Data

We take the most straightforward approach to evaluate data efficiency. We start with only $1\%$ of the original training data. We always use the full amount of testing data. We then gradually increase the amount of training data to $2\%$, $4\%$, $8\%$, $10\%$, and $20\%$ of the original to perform extensive multi-scale evaluation.

## 4 Experimental Setup

**Dataset** We evaluate our models on the CIFAR-10 and CIFAR-100 datasets (Krizhevsky, 2009). We use the same training, validation and testing data split ($45,000/5,000/10,000$) for all of our experiments. We also keep the same random seed for data preprocessing and augmentation. Therefore, we present apple to apple comparisons for all label representations.

**Speech Label Generation** We generate the speech labels shown in Figure 1 by following the standard practice as in recent works (Naranjo-Alcazar et al., 2019; Zhang et al., 2019):

- We first generate the English speech audio automatically with a text-to-speech (TTS)[1] system from the text labels in the corresponding dataset. Therefore, all the speech labels are pronounced consistently by the same API with the same parameters for controlled experiments. We leave the exploration of different languages and intonations as future work.
- We save each audio file in the WAVE format with the 16-bit pulse-code modulation encoding, and trim the silent edges from all audio files.
- Since different speech signals may last for various lengths, we preprocess[2] each speech label to generate a Log Mel spectrogram to maintain the same dimension. We use a sampling rate of $22,050$ Hz, $64$ Mel frequency bands, and a hop length of $256$. Another advantage of this preprocessing into spectrograms is that we can then utilize convolutional neural networks as the speech decoder to reconstruct our speech labels. Meanwhile, we convert the amplitudes to the decibel scale.
- Finally, the spectrograms are shaped into a $N \times N$ matrix with values ranging from $-80$ to $0$, where $N$ is double the dimension of the image input. Our resulting speech spectrogram can be viewed as a 2D image.

Given the first step in this procedure, note that for a given class (e.g. "bird") there is only one corresponding spectrogram. Therefore, the improved robustness we observe is not a result of any label-augmentation.

**Other Labels** For a deeper understanding of which properties of speech labels introduce different feature learning characteristics, we replicate all the experiments using the following high-dimensional label variants. We also show visualizations for all the label variants in Figure 1.

- *shuffled-speech* We cut the speech spectrogram image into $64$ slices along the time dimension, shuffle the order of them, and then combine them back together as an image. Although the new image cannot be converted back to a meaningful audio, it preserves the frequency information from the original audio. More importantly, this variant does not change the entropy or dimensionality of the original speech label.
- *constant-matrix* In contrast, the constant matrix represents another extreme of high-dimensional label representation, with all elements having the same value and zero entropy. The constant-matrix label has the same dimension as the speech label, and values are evenly spaced with a range of $80$ (which is also the range of the speech labels).
- *Gaussian-composition* We obtain this representation by plotting a composition of 2D Gaussians directly as images. Each composition is obtained by adding 10 Gaussians with uniformly sampled positions and orientations.
- *random/uniform-matrix* We also adopt a matrix with same dimensions as the above high-dimensional labels by randomly sampling from a uniform distribution. We additionally

---

[1] https://cloud.google.com/text-to-speech/
[2] https://librosa.github.io/

construct a uniform random vector with low dimensionality to inspect whether dimensionality matters. Throughout the paper, we will use random matrix and uniform matrix interchangeably to refer to the same representation.

- *BERT embedding* We obtain the BERT embedding from the last hidden state of a pre-trained BERT model (Devlin et al., 2018; Wolf et al., 2020). We remove outlines larger than twice the standard deviation and normalize the matrix to the same range of our other high-dimensional labels. The BERT embedding results in a $64 \times 64$ matrix.
- *GloVe embedding* Similarly, we directly use the pretrained GloVe (Pennington et al., 2014) word vectors.[3] This results in a 50-dimensional label vector.

**Models** We take three widely used convolutional networks as our base model: VGG19 (Simonyan & Zisserman, 2014), ResNet-32 and ResNet-110 (He et al., 2015). Here we define that a categorical classification model has two parts: an image encoder $I_e$ and a category (text) decoder $I_{td}$. Traditionally, $I_{td}$ represents fully connected layers following various forms of convolutional backbones $I_e$. Finally, a softmax function is applied to the output from the last layer of $I_{td}$ to convert the predictions to a categorical probability distribution. The prediction is retrieved from the class with the highest probability.

Similarly, we define that the models for our high-dimensional labels consists of two parts: an image encoder and a label decoder $I_{ld}$. Throughout the paper, we use the same image encoder but replace the category decoder $I_{td}$ with one dense layer and several transpose convolutional layers as $I_{ld}$. All the layers inside $I_{ld}$ are equipped with batch normalization (Ioffe & Szegedy, 2015) and leaky ReLU with a 0.01 negative slope.

Overall, the majority of parameters of the network comes from the image encoder which is shared across both categorical labels and other label representations. The decoder for high-dimensional labels increases the number of parameters by a very limited amount (see Appendix for all the numbers). Thus, our experiments are well-controlled with respect to the number of parameters.

**Learning** We train the categorical model to minimize the traditional cross-entropy objective function. We minimize Equation 6 for other models with high-dimensional labels. We use the smooth L1 loss (Huber loss) $L_s$ shown in Equation 7. Here, $y_i$ is the predicted label matrix while $s_i$ stands for the ground-truth matrix.

$$\min_{\theta_s} \sum_i \mathcal{L}_s(y_i, s_i) \tag{6}$$

$$\mathcal{L}_s(y_i, s_i) = \begin{cases} \frac{1}{2}(y_i - s_i)^2, & \text{if } |y_i - s_i| \leq 1 \\ |y_i - s_i|, & \text{otherwise} \end{cases} \tag{7}$$

We optimize all the networks using stochastic gradient descent (SGD) with back-propagation, and we select the best model based on the accuracy on the validation set.

**Evaluation** For categorical models, a prediction is considered correct if the class with the highest probability indicates the same category as the target class. For high-dimensional labels, we provide two types of measurements. Given the model output, we select the ground-truth label that minimizes the distance to the output. We refer to this as the "nearest neighbor" (NN) choice. The other criteria is to check whether the smooth L1 loss is below a certain threshold. We use Amazon Mechanical Turk (Sorokin & Forsyth, 2008) to validate that generated speech below our threshold is correctly identifiable by humans. In our experiments, we find 3.5 is a reasonable threshold. The human evaluations on the speech label demonstrate that our metric captures both the numerical performance and the level of interpretability of the generated speech output. Note that we mainly rely on the NN method for evaluation and only refer to the threshold method to demonstrate the qualitative results from the speech label.

## 5 RESULTS AND DISCUSSION

### 5.1 DO ALL THE MODELS LEARN TO CLASSIFY IMAGES?

We report the classification accuracy for all the labels in Table 1. Speech labels, shuffled-speech and Gaussian-composition labels achieve comparable accuracy with the traditional categorical labels,

---

[3]https://nlp.stanford.edu/projects/glove/

while the constant-matrix performs slightly worse than these representations. This suggests that the constant-matrix label is harder to train with. We verify this observation by visualizing the training curves for all label representations on CIFAR-100 with the ResNet-110 image encoder in Figure 2. The training curves show that the constant-matrix model takes longer to converge than the others, and converges to a higher loss.

| Labels | CIFAR-10 Accuracy (%) | | | CIFAR-100 Accuracy (%) | | |
|---|---|---|---|---|---|---|
| | VGG19 | ResNet32 | ResNet110 | VGG19 | ResNet32 | ResNet110 |
| Category | $92.82 \pm 0.08$ | $92.34 \pm 0.25$ | $93.23 \pm 0.29$ | $70.98 \pm 0.10$ | $68.05 \pm 0.72$ | $70.03 \pm 0.49$ |
| Speech Threshold | $91.97 \pm 0.17$ | $91.90 \pm 0.04$ | $92.44 \pm 0.11$ | $69.13 \pm 0.75$ | $61.08 \pm 0.27$ | $67.88 \pm 0.16$ |
| Speech NN | $92.12 \pm 0.18$ | $92.34 \pm 0.01$ | $92.73 \pm 0.08$ | $70.27 \pm 0.61$ | $64.74 \pm 0.36$ | $69.51 \pm 0.25$ |
| Shuffle Threshold | $92.27 \pm 0.16$ | $91.49 \pm 0.22$ | $92.44 \pm 0.05$ | $67.04 \pm 0.41$ | $51.95 \pm 0.85$ | $63.00 \pm 1.45$ |
| Shuffle NN | $92.64 \pm 0.19$ | $92.72 \pm 0.20$ | $92.92 \pm 0.24$ | $70.88 \pm 0.31$ | $64.23 \pm 0.80$ | $69.41 \pm 0.66$ |
| Composition Threshold | $91.53 \pm 0.24$ | $91.49 \pm 0.22$ | $92.36 \pm 0.02$ | $68.06 \pm 0.28$ | $60.54 \pm 0.54$ | $67.17 \pm 0.40$ |
| Composition NN | $91.94 \pm 0.22$ | $92.39 \pm 0.17$ | $93.07 \pm 0.03$ | $70.20 \pm 0.23$ | $66.72 \pm 0.44$ | $70.62 \pm 0.20$ |
| Constant Threshold | $88.27 \pm 0.63$ | $88.50 \pm 0.25$ | $89.27 \pm 0.15$ | $62.99 \pm 0.11$ | $55.70 \pm 0.36$ | $58.78 \pm 0.18$ |
| Constant NN | $88.33 \pm 0.65$ | $88.61 \pm 0.27$ | $89.37 \pm 0.13$ | $34.29 \pm 2.40$ | $19.00 \pm 1.19$ | $24.46 \pm 3.70$ |

Table 1: Classification accuracy on CIFAR-10 and CIFAR-100 for all label representations. Speech labels, shuffled speech labels, and composition of Gaussian labels all achieve comparable accuracies with categorical labels. Constant matrix labels perform slightly worse than the others.

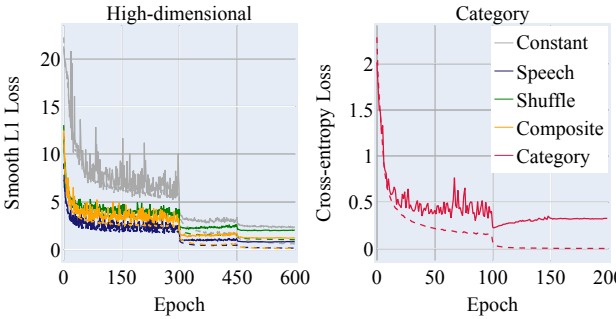

Figure 2: Training and validation losses on CIFAR-10 dataset with ResNet-110 image encoder for models with the speech / shuffled speech / composition of Gaussians / constant matrix labels (left) and categorical labels (right). All of the models are trained to converge. The model trained with constant matrix labels converges slower than models trained with other high dimensional labels.

## 5.2 FEATURE ROBUSTNESS

In order to evaluate the robustness of the models, we take all the trained models (Table 1) as the starting points for adversarial attacks with the FGSM and the iterative method. We apply an $\epsilon$ from 0 to 0.3 with a 0.05 increment to the normalized images by following the original FGSM setup (Goodfellow et al., 2014) and test the model accuracy for each $\epsilon$ value. We only run attacks on the original correctly classified images and mark the original misclassified images as incorrect when we compute the accuracy for all $\epsilon$ values. We provide the accuracy computed on the subset of test images that are initially correctly classified in the Appendix, though the ranking among different models remains the same.

Figure 3 shows the test accuracy under various attacks. Although the accuracy of all models decreases as the attack becomes stronger (larger $\epsilon$), the models with speech labels, shuffled speech labels, and composition of Gaussian labels perform consistently much better than the models with traditional categorical labels across all three image encoders, for all types of attacks, and on both CIFAR datasets. Uniform random matrix labels perform similarly well in this setting (see the Appendix for details). Interestingly, models with constant matrix labels perform worse than all other models with high-dimensional labels, suggesting that there are some inherent properties that enhance model robustness beyond simply high dimensionality.

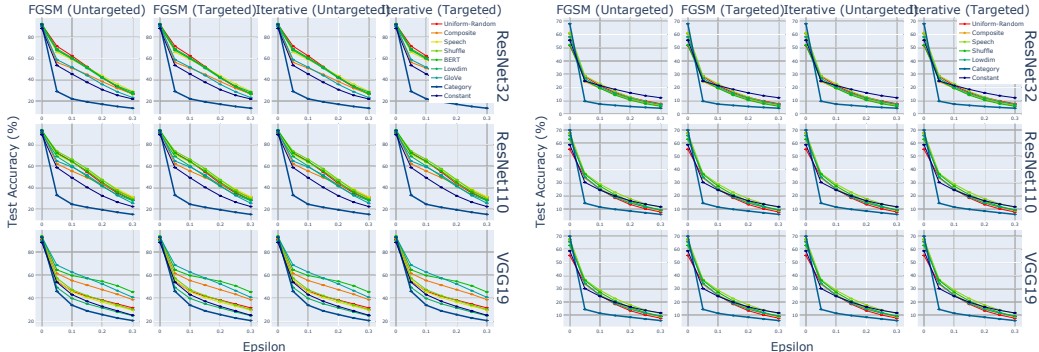

Figure 3: Test accuracy under adversarial attacks on CIFAR-10 (left four columns) and CIFAR-100 (right four columns). The accuracy evaluated by the threshold and the nearest neighbor is plotted in solid and dotted lines respectively. We show the results of targeted and untargeted FGSM and iterative method on three image encoders with three random seeds. The horizontal axis indicates the strength of different attacks.

## 5.3 FEATURE EFFECTIVENESS

With the CIFAR-10 dataset, we train models with various label representations using 1%, 2%, 4%, 8%, 10%, and 20% of the training data. For each amount of data, we train with the VGG19, ResNet-32, and ResNet-110 image encoders with five different random seeds. To conduct controlled experiments, we use the exact same training procedures and hyperparameters as the full-data experiments, so that the only difference is the amount of training data. All models are evaluated on the same validation set and test set. Figure 4 reports the test accuracy. Similar to the results from the robustness evaluation, speech labels, shuffled speech labels, composition of Gaussian labels, and uniform random labels achieve higher accuracies than the models with categorical labels for both VGG19 and ResNet-110, and comparable results for ResNet-32. The results demonstrate that the speech models are able to learn more generalizable and effective features with less data. This property is extremely valuable when the amount of training data is limited.

Additionally, our results suggest that label-smoothing does not provide further benefits when the amount of training data is limited, as discussed above. Lastly, the performance of the models trained on constant matrix labels is consistent with that in the robustness experiment: it performs worse than all other high-dimensional labels. We provide further analysis in the next section.

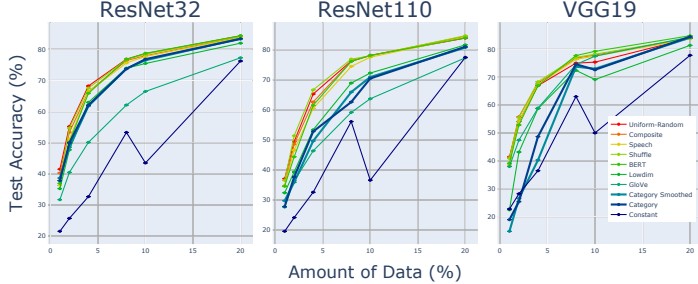

Figure 4: Test accuracy when limited training data is available. Accuracy is computed using the nearest-neighbor method

## 5.4 WHAT IS SPECIAL ABOUT AUDIO LABELS?

Our experiments for robustness and data efficiency suggest that high-dimensional labels hold some interesting inherent property beyond just high-dimensionality that encourage learning of more robust

and effective features. We hypothesize that high-dimensional label representations with high entropy provide stronger learning signals which give rise to better feature representations.

To verify our hypothesis, we measure several standard statistics over various label representations, shown in Table 2. Specifically, we measure the normalized L1 and L2 distance between pairs of labels for each representation. We further measure the entropy for each individual label.

| Label Types | CIFAR-10 | | | CIFAR-100 | | |
|---|---|---|---|---|---|---|
| | **Entropy** | L1 Distance | L2 Distance | **Entropy** | L1 Distance | L2 Distance |
| Category | $0.47 \pm 0.00$ | $2.00 \pm 0.00$ | $1.41 \pm 0.00$ | $0.08 \pm 0.00$ | $2.00 \pm 0.00$ | $1.41 \pm 0.00$ |
| Constant | $0.00 \pm 0.00$ | $26.07 \pm 15.72$ | $26.07 \pm 15.72$ | $0.00 \pm 0.00$ | $21.76 \pm 15.16$ | $21.76 \pm 15.16$ |
| Speech | $\mathbf{11.35} \pm 0.35$ | $23.80 \pm 5.16$ | $12.95 \pm 2.70$ | $\mathbf{11.37} \pm 0.32$ | $21.45 \pm 5.53$ | $13.15 \pm 2.19$ |
| Shuffle | $\mathbf{11.35} \pm 0.35$ | $35.29 \pm 2.18$ | $18.87 \pm 1.53$ | $\mathbf{11.37} \pm 0.32$ | $34.40 \pm 2.59$ | $17.81 \pm 1.24$ |
| Composite | $\mathbf{12.00} \pm 0.00$ | $24.13 \pm 3.36$ | $19.41 \pm 1.47$ | $\mathbf{12.00} \pm 0.00$ | $25.75 \pm 4.72$ | $20.60 \pm 2.96$ |
| BERT | $\mathbf{11.17} \pm 0.00$ | $5.71 \pm 0.94$ | $2.06 \pm 0.24$ | $\mathbf{11.17} \pm 0.00$ | $7.89 \pm 2.84$ | $2.63 \pm 0.70$ |
| GloVe | $\mathbf{5.64} \pm 0.00$ | $7.35 \pm 1.69$ | $1.30 \pm 0.31$ | $\mathbf{5.64} \pm 0.00$ | $5.62 \pm 0.90$ | $0.99 \pm 0.16$ |

Table 2: Different basic statistics of all types of label representations. Labels that encourage more robust and effective feature learning also have higher entropy than other label forms.

Interestingly, although the Manhattan and Euclidean distance between pairs of labels do not show any particularly useful patterns, the average entropy of the speech labels, the shuffled speech labels, and composition of Gaussian labels are all higher than that of the constant matrix and original categorical labels. The ranking of the entropy between these two groups exactly matches the performance in our robustness and data efficiency experiments shown in Figure 3 and Figure 4. This correlation suggests that high dimensional labels with high entropy may have positive impacts on robustness and data-efficient training.

We further validate that the benefits come from both high dimensionality and high entropy by training a model with low-dimensional and high-entropy labels. We generated these labels by sampling from a uniform distribution, following the same procedure as the uniform-random matrix label described previously. We found that while models trained with this label perform comparably to ones trained with high-dimensional high-entropy labels in terms of adversarial robustness (see Appendix), high-dimensional and high-entropy label models outperform the low-dimensional high-entropy model in terms of data efficiency, as shown by the curve "Low-dim" in Figure 4. We find a similar result for categorical models trained with label smoothing, which has been previously shown to improve adversarial robustness (Goibert & Dohmatob, 2020). In fact, high dimensionality is a prerequisite for high entropy because the maximum entropy is limited by the dimensionality of the label.

Note that the model trained with label smoothing uses the standard cross-entropy loss, meanwhile the low-dimensional high-entropy model is trained with the Huber loss. As a result, we argue that the loss is not responsible for the improved performance of models trained with high-dimensional, high-entropy labels.

## 5.5 VISUALIZATIONS

Throughout the training process, we visualized the learned features immediately after the image encoder layer of ResNet-110 with t-SNE (van der Maaten & Hinton, 2008), both for audio and categorical models on CIFAR-10 test set. The results are shown in Figure 5. We observe that the embedding of the learned features evolve as the training progresses. Compared with the feature embedding of the categorical model, the embedding of the speech models shows the formation of clusters at earlier stages of training. More separated clusters are also obtained towards convergence. We provide further visualizations and Grad-CAM interpretations in the Appendix.

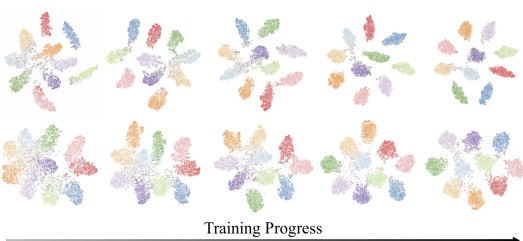

Figure 5: T-SNE progression for speech (top row) and categorical (bottom row) models with ResNet-110 image encoder. From left to right, the plot shows 10%, 30%, 50%, 70%, and 100% progress in training. The speech model develops distinctive clusters at an earlier stage and has better separated clusters overall.

## 6 CONCLUSION

We introduce a novel paradigm for the traditional image classification task by replacing categorical labels with high-dimensional, high-entropy matrices such as speech spectrograms. We demonstrate comparable accuracy on the original task with our speech labels, however, models trained with our speech labels achieve superior performance under various adversarial attacks and are able to learn in a more data efficient manner with only a small percentage of training data. We further study the inherent properties of high dimensional label representations that potentially introduce the advantages. Through a large scale systematic study on various label representations, we suggest that high-entropy, high-dimensional labels generally lead to more robust and data efficient training. Our work provides novel insights for the role of label representation in training deep neural networks.

### ACKNOWLEDGMENTS

This research is supported by NSF NRI 1925157 and DARPA MTO grant L2M Program HR0011-18-2-0020.

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

# A  APPENDIX

## A.1  THRESHOLD VALIDATION

We deployed a large-scale study on Amazon Mechanical Turk[4] to validate our choice of 3.5 as a classification threshold for the speech model.

In particular, we asked workers to listen to the outputs of the speech model and choose from the set of classes (with a "None" class for unintelligible output) the one which best fits the output. We assigned 3 workers to evaluate each output from the VGG19 speech model on CIFAR-10 test set. We chose a restrictive selection criteria to ensure maximum quality of responses. Only workers with a $\geq 99\%$ approval rating and at least $10,000$ approvals were selected.

To measure agreement between humans and our speech model, for each sample in the test set we determine the decision made by the model using our pre-selected threshold (loss $< 3.5$ is correct, while loss $\geq 3.5$ is incorrect). Then we compare these decisions to those of the human workers. When we count each of the three workers independently, we find that humans agree with the model $99.4\%$ of the time. When we take a majority vote (2/3 humans agreeing) we find that humans agree with the model $99.8\%$ of the time. We conclude that 3.5 is a reasonable threshold for evaluating the model.

## A.2  SUBSET ROBUSTNESS

Additional robustness evaluation is computed using the subset of the test images that are initially correctly classified by the models without any adversarial attacks (Figure 6). All test accuracy starts at $100\%$ and decreases with stronger attacks. The strength of the attacks is described by the value of epsilon.

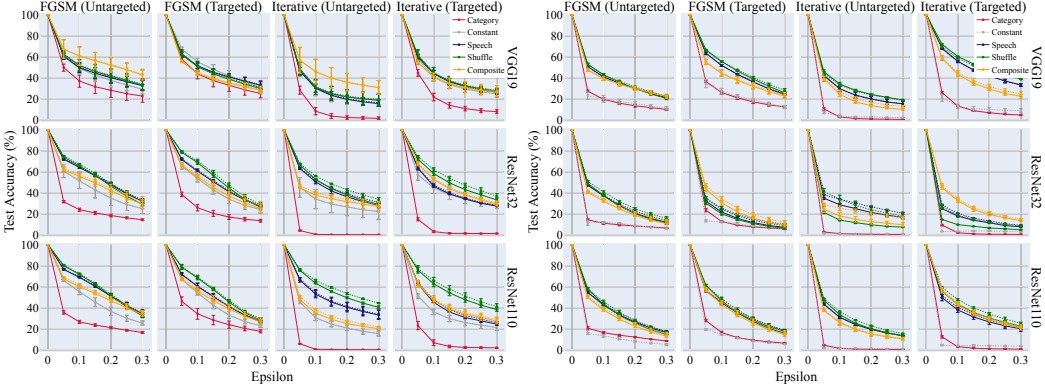

Figure 6: Test accuracy under adversarial attacks on CIFAR-10 (left four columns) and CIFAR-100 (right four columns) for the initially correct subset of the test images. The accuracy evaluated by the threshold and the nearest neighbor is plotted in solid and dotted lines respectively. The general trend from the subset is similar to that from the full test set.

## A.3  ADDITIONAL RESULTS ON ROBUSTNESS EVALUATION

We here include the full results on robustness evaluation on CIFAR-10 dataset in Figure 7.

---

[4]https://www.mturk.com

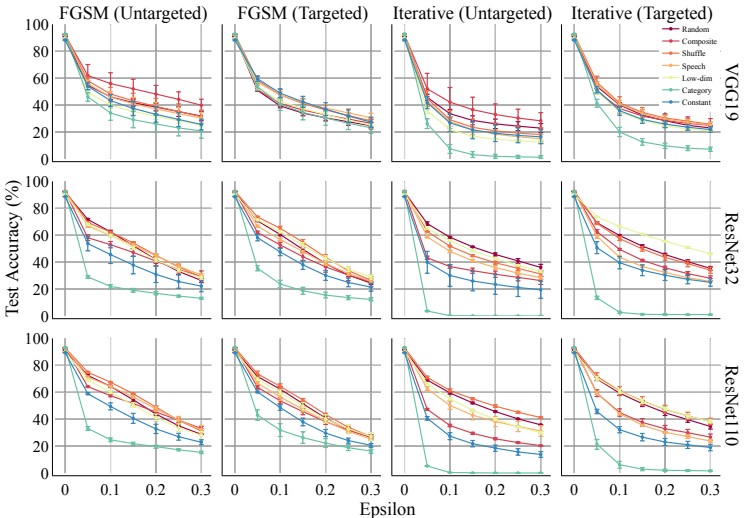

Figure 7: Full results of the robustness evaluation on CIFAR-10

## A.4 HYPERPARAMETERS

### A.4.1 IMPLEMENTATION DETAILS

We train the categorical models for 200 epochs with a starting learning rate of $0.1$, and decay the learning rate by $0.1$ at epoch 100 and 150. The high-dimensional models are trained for 600 epochs with the same initial learning rate, and we drop the learning rate by $0.1$ at epoch 300 and 450. All models are trained with a batch size of 128 using the SGD optimizer with $0.9$ momentum and $0.0001$ weight decay. One exception is when train categorical models with the VGG19 image encoder, we use a larger weight decay, $0.0005$. We implement our models using PyTorch Paszke et al. (2019). All experiments are performed on a single GeForce RTX 2080 Ti GPU. The limited-data experiments are conducted using the same settings as the full data experiments.

### A.4.2 ARCHITECTURE PARAMETERS

We provide the parameter counts for the all models in Table 3 and Table 4. The majority of the parameters come from the image encoders. High-dimensional models have slightly more parameters than categorical models due to the high-dimensional label decoder (Table 5).

| Model | CIFAR-10 | | |
|---|---|---|---|
| | VGG19 | ResNet32 | ResNet110 |
| Category | $2 \times 10^7$ | $4.67 \times 10^5$ | $1.73 \times 10^6$ |
| High-dimensional | $2.01 \times 10^7$ | $5.80 \times 10^5$ | $1,84 \times 10^6$ |

Table 3: Total number of parameters of the category and high-dimensional models for CIFAR-10 dataset

## A.5 DATASET

To demonstrate the effectiveness of our proposed method, we evaluate our models on the CIFAR-10 and CIFAR-100 datasets Krizhevsky (2009). For each dataset, we train different models on the same training set and evaluate the models on the same validation set using the same random seeds for fair comparisons. To preprocess the training images, we randomly crop them with a padding size 4 and perform random horizontal flips. All CIFAR images are normalized with mean $(0.4914, 0.4822, 0.4465)$ and standard deviation $(0.2023, 0.1994, 0.2010)$ of the training set.

| Model | CIFAR-100 | | |
|---|---|---|---|
| | VGG19 | ResNet32 | ResNet110 |
| Category | $2.01 \times 10^7$ | $4.73 \times 10^5$ | $1.74 \times 10^5$ |
| High-dimensional | $2.02 \times 10^7$ | $5.80 \times 10^5$ | $1.84 \times 10^5$ |

Table 4: Total number of parameters of the category and high-dimensional models for CIFAR-100 dataset

| Layer | Input | Output | Kernel | Stride | Padding |
|---|---|---|---|---|---|
| Dense | $I_e$ out | 64 | - | - | - |
| ConvTranspose 2D | $64 \times 1 \times 1$ | $64 \times 4 \times 4$ | $4 \times 4$ | $1 \times 1$ | 0 |
| ConvTranspose 2D | $64 \times 4 \times 4$ | $32 \times 8 \times 8$ | $4 \times 4$ | $2 \times 2$ | $1 \times 1$ |
| ConvTranspose 2D | $32 \times 8 \times 8$ | $16 \times 16 \times 16$ | $4 \times 4$ | $2 \times 2$ | $1 \times 1$ |
| ConvTranspose 2D | $16 \times 16 \times 16$ | $8 \times 32 \times 32$ | $4 \times 4$ | $2 \times 2$ | $1 \times 1$ |
| ConvTranspose 2D | $8 \times 32 \times 32$ | $1 \times 64 \times 64$ | $4 \times 4$ | $2 \times 2$ | $1 \times 1$ |

Table 5: The architecture of the high-dimensional label decoder $I_{ld}$. The input dimension of the first dense layer is the dimension of the output of the image encoder $I_e$. The output of the last ConvTranspose2d layer is the target label.

**CIFAR-10** consists of $60,000$ images of size $32 \times 32$ uniformly distributed across 10 classes. The dataset comes with $50,000$ training images and $10,000$ test images. We use a $45,000/5,000$ training/validation split.

**CIFAR-100** also comprises $60,000$ images of size $32 \times 32$, but it has 100 classes each containing 600 images. The dataset is split into $50,000$ training images and $10,000$ test images. We randomly select $5,000$ images from the training images to form the validation set.

A.6   VISUALIZATIONS

In addition to the progressive t-SNE plots we presented in the main context, we plot the embedding of all three types of image encoders of models trained on speech labels, categorical labels, and constant matrix labels in Figure 8. We only show the results from the models with highest accuracy. Similarly, we observe that speech models have better separation of clusters. The feature embedding of the constant model is worse than that of the speech model, further confirming that the speech representation is unique in addition to its dimensionality.

**Grad-CAM** We visualize activations from beginning, intermediate, and final layers in the image encoders for both speech and categorical models in Figure 9. We see that the input activations for the speech model conform more tightly than those of the categorical model to the central object of each image at all three stages. This may suggest that the speech model learns more discriminative features than the categorical model. These features are also visually more easily understandable to humans.

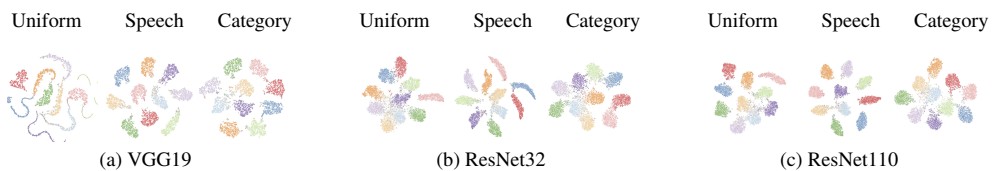

Figure 8: T-SNE of the best uniform (left), speech (middle), and categorical (right) models trained with the same random seed. Speech models show best separated clusters across all three models.

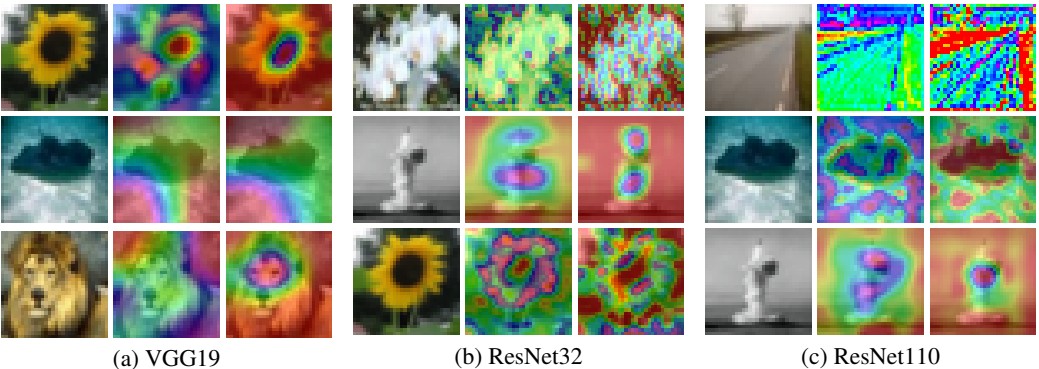

(a) VGG19  (b) ResNet32  (c) ResNet110

Figure 9: Grad-CAM visualization of learned features. Each triplet contains (from left to right) an unaltered image, a categorical model visualization, and a speech model visualization. From top to bottom, the activations are taken from some beginning, intermediate, and final layer in the respective image encoders. Speech models learn more discriminative features than categorical models.

