# OpenReview forum: "Beyond Categorical Label Representations for Image Classification"
_ICLR.cc/2021/Conference — ICLR 2021 Poster_

### Official Review · AnonReviewer2 · 2020-10-28
**Interesting proposed method but not enough theoretical support.**

**Rating:** 4
**Confidence:** 4

**Review:**

This work suggests an alternative training method where the ground truth label is given as speech signal. The authors hypothesize the labels with high-dimensionality & high entropy will help the network learn better feature representation. The hypothesis is empirically examined by the authors using commonly used CIFAR-10 and CIFAR-100 datasets. The experiment results show that the proposed approach is effective in terms of adversarial robustness and data efficiency.

Strength:
The idea of giving speech signal as a proxy for categorical label is interesting.

Weakness:  Unclear source of performance for data efficiency and robustness on adversarial attack

The proposed claim in this work seems quite bold and is mostly built on empirical observation.  Also, I think the proposed approach is somewhat similar to the popular multi-modal representation learning schemes, which is quite common these days. The main difference is that the authors train the network by targeting speech signals itself. I recommend the authors to compare the proposed method with previous multi-modal learning training schemes such as [1, 2] and many more.

Ratings:

Although the paper shows some interesting research direction, the way the authors show the effectiveness of the proposed method is mostly built on empirical results, which is not enough to claim such a bold argument. For this reason, I recommend rejection.

References

[1] Look, Listen and Learn (https://arxiv.org/abs/1705.08168)

[2] Objects that Sound (https://arxiv.org/abs/1712.06651)

---

> ### Author Response · Authors · 2020-11-13
> **Response to Reviewer 2**
>
> Thank you for your constructive feedback. We hope that we can clarify and address your concerns with the point response below.
>
> - “The proposed claim in this work seems quite bold and is mostly built on empirical observation”
>
> We argue that empirical observation is an important research practice in deep learning and representation learning which has historically been responsible for many major milestones. We have made sure to provide strong evidence through thorough and systematic experiments. Moreover, since the direction we are studying in this paper is relatively underexplored, we believe that our research can serve as an important reference for future theoretical work.
>
> - “I think the proposed approach is somewhat similar to the popular multi-modal representation learning schemes, which is quite common these days. The main difference is that the authors train the network by targeting speech signals itself. I recommend the authors to compare the proposed method with previous multi-modal learning training schemes such as [1, 2] and many more.”
>
> We will add a section in our related works to discuss multi-modal representation learning. Our focus and training scheme are not directly comparable with multi-modal representation learning:
>
> (1) The label representations used in our paper are very different from the data used in multi-modal learning such as captions, sound, and videos. The extra modalities used in multi-modal learning mostly come with rich extra information that cannot be directly extracted from the image or video representation. However, our contribution is to suggest that underexplored label representations, especially “high-dimensional high-entropy” label representations can serve as strong alternatives to traditional categorical labels with more robust features and more efficient learning in the image classification setting. In addition to speech labels, our paper also covers uniform-random matrix, shuffled speech label, composition of Gaussian patterns, constant matrix, and categorical label with label smoothing.
>
> (2) The focus is also different. The recommended works from the reviewer aim at learning the correspondence between a video and audio clip in a self-supervised manner. Specifically, the main idea is to train a network to distinguish if the given video and audio clip is well corresponded. Both the video and audio are input. The learned feature representation can then be used to train other linear classifiers on downstream tasks such as sound classification, acoustic scene classification, and image classification. The learning paradigm falls well into the visual-audio correspondence self-supervised learning.
>
> However, our approach does not aim at self-supervised learning and multi-modal learning. Our task is the traditional image classification task. Instead of learning effective feature representations for the downstream task, our goal is to study underexplored label representations and their effectiveness. The speech modality only serves as a variant of the categorical label representation. Other label representations in our paper do not count a different modality. Importantly, the input of our network only contains the raw images without other modalities.
>
> (3) The training procedure differs from our paper and the works recommended by the reviewer. Our training procedure follows the standard supervised learning image classification task. Given an image input, the network is optimized to generate the correct label. In the works recommended by the reviewer, a network is first pre-trained on a predefined pretext task. A second network is then trained in a supervised manner on a different downstream task with the learned features from the first network. Since our approach does not assume a pretext task or a pre-training procedure, the training methods are not directly comparable.
>
> Please kindly let us know if we address your concerns. Thank you!

---

> ### Author Response · Authors · 2020-11-20
> **Responses to Reviewer 2 (Round 2 Rebuttal)**
>
> Dear Reviewer 2,
>
> As the second round of rebuttal begins today, please kindly let us know if we have addressed your concerns. We would love to try our best to address your concerns, so please feel free to let us know. Thank you!

---

### Official Review · AnonReviewer1 · 2020-10-29
**Interesting observation, nice experiments**

**Rating:** 7
**Confidence:** 3

**Review:**

The authors study the effect of data labels on the quality of trained models. More specifically, the authors use audio labels rather than traditional categorical probabilities for model training and get surprising and interesting results.  The results show that high dimensional and high entropy label representations are more useful, which is observed in the experiments related to robustness and a limited amount of training data. Such a result is very interesting and suggests that the label representation can be further explored and potentially plan an important role.

This paper starts an interesting direction and conducts nice experiments. The paper is easy to follow and well written. The hypothesis about the relationship between high entropy and training effectiveness is also a good observation.

I am also interested in that if the audio signal is replaced by pre-trained embeddings, like glove or BERT, as label representation, how the effectiveness of labels is compared with the audio signals?

---

> ### Author Response · Authors · 2020-11-13
> **Reponses to Reviewer 1 (Part 3)**
>
> ##### BERT Attack Results:
>
> | ResNet32                |  Epsilon 0  | Epsilon 0.05 | Epsilon 0.1 | Epsilon 0.15 | Epsilon 0.2 | Epsilon 0.25 | Epsilon 0.3 |
> |-------------------------|:-----------:|:------------:|:-----------:|:------------:|:-----------:|:------------:|:-----------:|
> | FSGM MEAN               | 0.927733333 |  0.690766667 | 0.607433333 |  0.51736667  |  0.42313333 |  0.33553333  |  0.26746667 |
> | FSGM STD                | 0.001625833 |  0.015631165 |  0.0345558  |  0.04923437  |  0.05716374 |  0.05364516  |  0.04588489 |
> | FSGM Targeted MEAN      | 0.927733333 |  0.738366667 | 0.670866667 |    0.5776    |  0.46483333 |  0.36613333  |    0.2873   |
> | FSGM Targeted STD       | 0.001625833 |  0.012159907 | 0.025550408 |  0.04388633  |  0.05841955 |  0.05822837  |  0.0491692  |
> | Iterative MEAN          | 0.927733333 |  0.616066667 |    0.4877   |  0.40496667  |  0.34303333 |    0.2986    |  0.25896667 |
> | Iterative STD           | 0.001625833 |  0.029737574 | 0.048071925 |  0.05809392  |  0.06387256 |  0.06849299  |   0.066176  |
> | Iterative Targeted MEAN | 0.927733333 |  0.711233333 | 0.631633333 |  0.57116667  |  0.51126667 |  0.46513333  |    0.4242   |
> | Iterative Targeted STD  | 0.001625833 |  0.02125049  | 0.035287722 |  0.04435903  |  0.05630385 |   0.0603089  |  0.0656461  |
>
> | VGG19                   |  Epsilon 0  | Epsilon 0.05 | Epsilon 0.1 | Epsilon 0.15 | Epsilon 0.2 | Epsilon 0.25 | Epsilon 0.3 |
> |-------------------------|:-----------:|:------------:|:-----------:|:------------:|:-----------:|:------------:|:-----------:|
> | FSGM MEAN               | 0.936866667 |    0.6461    |    0.5969   |  0.57206667  |  0.54673333 |  0.50866667  |  0.45506667 |
> | FSGM STD                | 0.000896289 |  0.006657327 | 0.003534119 |  0.00591383  |  0.01069268 |  0.00884892  |  0.00123423 |
> | FSGM Targeted MEAN      | 0.936866667 |    0.7262    | 0.684433333 |  0.65903333  |  0.63113333 |    0.5858    |  0.52106667 |
> | FSGM Targeted STD       | 0.000896289 |  0.012247857 | 0.006572924 |  0.01130059  |  0.0169512  |  0.01455026  |  0.01020016 |
> | Iterative MEAN          | 0.936866667 |  0.564933333 | 0.462633333 |  0.40366667  |    0.3629   |  0.33546667  |    0.317    |
> | Iterative STD           | 0.000896289 |  0.00280238  | 0.016914589 |  0.02437875  |  0.02829629 |  0.02835918  |  0.0265684  |
> | Iterative Targeted MEAN | 0.936866667 |  0.734033333 |    0.6511   |  0.58886667  |  0.54796667 |  0.51586667  |  0.48706667 |
> | Iterative Targeted STD  | 0.000896289 |  0.011402777 | 0.010247439 |   0.0160076  |  0.01668422 |  0.01494468  |  0.01494869 |
>
> ##### Data Efficiency
>
> | GloVe          |      1%     |      2%     |      4%     |     8%     |     10%    |     20%    |
> |----------------|:-----------:|:-----------:|:-----------:|:----------:|:----------:|:----------:|
> | ResNet32 MEAN  |    0.3141   |   0.40458   |   0.50112   |   0.62104  |   0.66468  |   0.77284  |
> | ResNet32 STD   | 0.017050161 | 0.011675855 | 0.010050552 | 0.00893501 | 0.00603569 | 0.00620729 |
> | ResNet110 MEAN |   0.27674   |    0.3636   |   0.46344   |   0.59132  |   0.6368   |   0.77408  |
> | ResNet110 STD  | 0.025722877 | 0.015070103 |  0.02634104 |  0.0180268 | 0.00697567 | 0.00309283 |
> | VGG19 MEAN     |   0.37956   |   0.47456   |    0.5894   |   0.7458   |   0.7759   |   0.84446  |
> | VGG19 STD      | 0.005315111 | 0.011771763 | 0.009727692 | 0.00290586 | 0.00634161 | 0.00480816 |

---

> > ### Comment · AnonReviewer1 · 2020-11-16
> > **Great experiments**
> >
> > Thanks for your feedback. My concerns are handled well.  How to represent our learning goal is a pretty interesting direction.

---

> > > ### Author Response · Authors · 2020-11-16
> > > **Thank you!**
> > >
> > > Thank you for your response! Please feel free to let us know if you have other questions in the future.

---

> ### Author Response · Authors · 2020-11-13
> **Reponses to Reviewer 1 (Part 2)**
>
> ##### GloVe Attack Results:
>
> | ResNet32                |  Epsilon 0  | Epsilon 0.05 | Epsilon 0.1 | Epsilon 0.15 | Epsilon 0.2 | Epsilon 0.25 | Epsilon 0.3 |
> |-------------------------|:-----------:|:------------:|:-----------:|:------------:|:-----------:|:------------:|:-----------:|
> | FSGM MEAN               | 0.908366667 |    0.5895    |    0.5178   |  0.43976667  |  0.35593333 |  0.28626667  |  0.23183333 |
> | FSGM STD                | 0.000971253 |  0.011358257 | 0.017911728 |  0.02619338  |  0.03930755 |  0.04576869  |  0.04143336 |
> | FSGM Targeted MEAN      | 0.908366667 |  0.663833333 |    0.5728   |  0.47496667  |  0.37633333 |  0.29713333  |    0.2388   |
> | FSGM Targeted STD       | 0.000971253 |  0.004045162 | 0.016937237 |  0.03755001  |  0.0529005  |  0.05715263  |  0.05345849 |
> | Iterative MEAN          | 0.908366667 |  0.536733333 | 0.406266667 |    0.3248    |  0.26723333 |  0.22213333  |  0.19023333 |
> | Iterative STD           | 0.000971253 |  0.005478443 |  0.00546382 |   0.0051643  |  0.01362583 |  0.01503806  |  0.02038929 |
> | Iterative Targeted MEAN | 0.908366667 |  0.654166667 | 0.558166667 |    0.4847    |    0.4289   |    0.3854    |    0.3449   |
> | Iterative Targeted STD  | 0.000971253 |  0.005201282 | 0.011447416 |  0.01825514  |  0.02280066 |   0.0317572  |  0.03139379 |
>
> | VGG19                   |  Epsilon 0  | Epsilon 0.05 | Epsilon 0.1 | Epsilon 0.15 | Epsilon 0.2 | Epsilon 0.25 | Epsilon 0.3 |
> |-------------------------|:-----------:|:------------:|:-----------:|:------------:|:-----------:|:------------:|:-----------:|
> | FSGM MEAN               | 0.927566667 |    0.6875    | 0.626433333 |  0.57446667  |  0.52426667 |  0.46656667  |  0.40953333 |
> | FSGM STD                | 0.001650253 |  0.013865425 | 0.026476468 |  0.03488572  |  0.04250604 |   0.0445426  |  0.04344702 |
> | FSGM Targeted MEAN      | 0.927566667 |  0.719066667 |    0.6587   |    0.6042    |  0.54856667 |  0.48763333  |  0.42463333 |
> | FSGM Targeted STD       | 0.001650253 |  0.014600457 | 0.032930837 |  0.03766656  |  0.04145001 |  0.04438404  |  0.04252262 |
> | Iterative MEAN          | 0.927566667 |  0.618733333 | 0.500066667 |  0.42243333  |  0.36636667 |  0.32313333  |  0.29333333 |
> | Iterative STD           | 0.001650253 |  0.019620737 | 0.049166384 |  0.05432949  |  0.05828562 |  0.05640854  |  0.05413172 |
> | Iterative Targeted MEAN | 0.927566667 |  0.709766667 | 0.613233333 |    0.5412    |    0.4884   |  0.45046667  |  0.41603333 |
> | Iterative Targeted STD  | 0.001650253 |  0.020705152 | 0.045928459 |  0.05445062  |  0.06164033 |  0.05794259  |  0.05452654 |

---

> ### Author Response · Authors · 2020-11-13
> **Reponses to Reviewer 1 (Part 1)**
>
> Thank you for your suggestions and the appreciation of our work.
>
> - “I am also interested in that if the audio signal is replaced by pre-trained embeddings, like glove or BERT, as label representation, how the effectiveness of labels is compared with the audio signals?”
>
> We are happy to report our newest experiments with BERT and GloVe embedding representations on CIFAR10. Specifically, we find that GloVe embedding labels (50-dim vector) outperform categorical labels in both data efficiency and adversarial robustness. The performance of the GloVe labels is comparable to that of the low-dimensional label experiment (see Figures 3 and 4). We also find that BERT embeddings labels (48x48 matrix) outperform both categorical labels and GloVe/low-dimensional labels and achieve comparable performance with our other high-dimensional high-entropy labels. These results are consistent with the observations in our paper. Namely, that high dimensionality and high entropy both improve feature representations. GloVe labels are higher entropy than categorical labels, but lower dimensionality/entropy than BERT labels.  We will perform the full set of experiments with GloVe and BERT on other datasets and report them as they finish running.
>
> Please kindly let us know if we address your concerns. Thank you!

---

### Official Review · AnonReviewer3 · 2020-10-29
**Recommendation to Accept**

**Rating:** 7
**Confidence:** 4

**Review:**

##########################################################################
Summary:

This paper presents a novel approach/perspective for improving data efficiency and robustness, other than existing research progress achieved by scientists, from model, optimizer and data perspective.

In particular, it proposes to introduce high dimensional and high entropy label representations for group truth, to improve image classification performance from two practical matters --- Robustness and data efficiency, while achieving comparable accuracy to text labels as the standard representation.  To valid its findings, the authors develop designed a set of comprehensive experiments for evaluation and comparison purposes, while making the best effort to not introducing variations from other angles, such as keeping the same data for training and testing and introducing adversary information consistently among all labeling representations.

##########################################################################
Reasons for score:

Overall, I vote for accepting.
I like the idea of approaching image classification problem from a new angle that are not well explored yet.
My major concerns are:
1. The clarity of the study on the underline true set of characteristics that contribute to the improvement, from speech label, shuffled-speech label, Gaussian-composition label, besides high dimension and high entropy.
2. The logic behind the adversary image generation, target vs non-target.
3. And the evidence/thinking process behind the pre-selected threshold of 3.5

Hopefully the authors can address my concern in the rebuttal period.

##########################################################################
Pros:

1. The paper provides a novel perspective for classification performance improvement, rather than from data, algorithms, or optimizers perspective. It shows the importance of label for supervised learning problems, can also come from the ways that we represent them, just only label quality.  For me, this approach is new and potentially expend to other applications, besides image classification

2. The experiment design is also quite comprehensive as it covers all potential perspectives and variations.

3. This inspiration of the idea to me is also quite natural and understandable, as for most of us, when recognizing an image, we express it not just in writing and can also in speech format.


##########################################################################
Cons:

Although the proposed representations have shown better performance in image classification problem, with evidence to support its out-performed robustness to adversarial attack and data efficiency -- achieving comparable accuracy with less data in training, I would still suggest the authors to conduct the following studies to enhance the quality of the paper:

1. It could be valuable to future investigate the inherent property that contributes to the improvement, besides high dimensionality and high entropy.

2. What’s the performance with high dimensional and high entropy label representations, comparing to text label, for other kind of the classification problems, such as NLP problems.

3. For speech label, in model evaluation, what’s the performance for the model, if we choose a speech to text process to obtain its ground truth, beside the two approaches mentioned in the paper -- “nearest neighbor” and a validated loss threshold.



##########################################################################
Questions during rebuttal period:

Please address and clarify the cons above

---

> ### Author Response · Authors · 2020-11-13
> **Responses to Reveiwer 3 (Part 4)**
>
> ##### BERT Attack Results:
>
> | ResNet32                |  Epsilon 0  | Epsilon 0.05 | Epsilon 0.1 | Epsilon 0.15 | Epsilon 0.2 | Epsilon 0.25 | Epsilon 0.3 |
> |-------------------------|:-----------:|:------------:|:-----------:|:------------:|:-----------:|:------------:|:-----------:|
> | FSGM MEAN               | 0.927733333 |  0.690766667 | 0.607433333 |  0.51736667  |  0.42313333 |  0.33553333  |  0.26746667 |
> | FSGM STD                | 0.001625833 |  0.015631165 |  0.0345558  |  0.04923437  |  0.05716374 |  0.05364516  |  0.04588489 |
> | FSGM Targeted MEAN      | 0.927733333 |  0.738366667 | 0.670866667 |    0.5776    |  0.46483333 |  0.36613333  |    0.2873   |
> | FSGM Targeted STD       | 0.001625833 |  0.012159907 | 0.025550408 |  0.04388633  |  0.05841955 |  0.05822837  |  0.0491692  |
> | Iterative MEAN          | 0.927733333 |  0.616066667 |    0.4877   |  0.40496667  |  0.34303333 |    0.2986    |  0.25896667 |
> | Iterative STD           | 0.001625833 |  0.029737574 | 0.048071925 |  0.05809392  |  0.06387256 |  0.06849299  |   0.066176  |
> | Iterative Targeted MEAN | 0.927733333 |  0.711233333 | 0.631633333 |  0.57116667  |  0.51126667 |  0.46513333  |    0.4242   |
> | Iterative Targeted STD  | 0.001625833 |  0.02125049  | 0.035287722 |  0.04435903  |  0.05630385 |   0.0603089  |  0.0656461  |
>
> | VGG19                   |  Epsilon 0  | Epsilon 0.05 | Epsilon 0.1 | Epsilon 0.15 | Epsilon 0.2 | Epsilon 0.25 | Epsilon 0.3 |
> |-------------------------|:-----------:|:------------:|:-----------:|:------------:|:-----------:|:------------:|:-----------:|
> | FSGM MEAN               | 0.936866667 |    0.6461    |    0.5969   |  0.57206667  |  0.54673333 |  0.50866667  |  0.45506667 |
> | FSGM STD                | 0.000896289 |  0.006657327 | 0.003534119 |  0.00591383  |  0.01069268 |  0.00884892  |  0.00123423 |
> | FSGM Targeted MEAN      | 0.936866667 |    0.7262    | 0.684433333 |  0.65903333  |  0.63113333 |    0.5858    |  0.52106667 |
> | FSGM Targeted STD       | 0.000896289 |  0.012247857 | 0.006572924 |  0.01130059  |  0.0169512  |  0.01455026  |  0.01020016 |
> | Iterative MEAN          | 0.936866667 |  0.564933333 | 0.462633333 |  0.40366667  |    0.3629   |  0.33546667  |    0.317    |
> | Iterative STD           | 0.000896289 |  0.00280238  | 0.016914589 |  0.02437875  |  0.02829629 |  0.02835918  |  0.0265684  |
> | Iterative Targeted MEAN | 0.936866667 |  0.734033333 |    0.6511   |  0.58886667  |  0.54796667 |  0.51586667  |  0.48706667 |
> | Iterative Targeted STD  | 0.000896289 |  0.011402777 | 0.010247439 |   0.0160076  |  0.01668422 |  0.01494468  |  0.01494869 |
>
> ##### Data Efficiency
>
> | GloVe          |      1%     |      2%     |      4%     |     8%     |     10%    |     20%    |
> |----------------|:-----------:|:-----------:|:-----------:|:----------:|:----------:|:----------:|
> | ResNet32 MEAN  |    0.3141   |   0.40458   |   0.50112   |   0.62104  |   0.66468  |   0.77284  |
> | ResNet32 STD   | 0.017050161 | 0.011675855 | 0.010050552 | 0.00893501 | 0.00603569 | 0.00620729 |
> | ResNet110 MEAN |   0.27674   |    0.3636   |   0.46344   |   0.59132  |   0.6368   |   0.77408  |
> | ResNet110 STD  | 0.025722877 | 0.015070103 |  0.02634104 |  0.0180268 | 0.00697567 | 0.00309283 |
> | VGG19 MEAN     |   0.37956   |   0.47456   |    0.5894   |   0.7458   |   0.7759   |   0.84446  |
> | VGG19 STD      | 0.005315111 | 0.011771763 | 0.009727692 | 0.00290586 | 0.00634161 | 0.00480816 |

---

> ### Author Response · Authors · 2020-11-13
> **Responses to Reveiwer 3 (Part 3)**
>
> ##### GloVe Attack Results:
>
> | ResNet32                |  Epsilon 0  | Epsilon 0.05 | Epsilon 0.1 | Epsilon 0.15 | Epsilon 0.2 | Epsilon 0.25 | Epsilon 0.3 |
> |-------------------------|:-----------:|:------------:|:-----------:|:------------:|:-----------:|:------------:|:-----------:|
> | FSGM MEAN               | 0.908366667 |    0.5895    |    0.5178   |  0.43976667  |  0.35593333 |  0.28626667  |  0.23183333 |
> | FSGM STD                | 0.000971253 |  0.011358257 | 0.017911728 |  0.02619338  |  0.03930755 |  0.04576869  |  0.04143336 |
> | FSGM Targeted MEAN      | 0.908366667 |  0.663833333 |    0.5728   |  0.47496667  |  0.37633333 |  0.29713333  |    0.2388   |
> | FSGM Targeted STD       | 0.000971253 |  0.004045162 | 0.016937237 |  0.03755001  |  0.0529005  |  0.05715263  |  0.05345849 |
> | Iterative MEAN          | 0.908366667 |  0.536733333 | 0.406266667 |    0.3248    |  0.26723333 |  0.22213333  |  0.19023333 |
> | Iterative STD           | 0.000971253 |  0.005478443 |  0.00546382 |   0.0051643  |  0.01362583 |  0.01503806  |  0.02038929 |
> | Iterative Targeted MEAN | 0.908366667 |  0.654166667 | 0.558166667 |    0.4847    |    0.4289   |    0.3854    |    0.3449   |
> | Iterative Targeted STD  | 0.000971253 |  0.005201282 | 0.011447416 |  0.01825514  |  0.02280066 |   0.0317572  |  0.03139379 |
>
> | VGG19                   |  Epsilon 0  | Epsilon 0.05 | Epsilon 0.1 | Epsilon 0.15 | Epsilon 0.2 | Epsilon 0.25 | Epsilon 0.3 |
> |-------------------------|:-----------:|:------------:|:-----------:|:------------:|:-----------:|:------------:|:-----------:|
> | FSGM MEAN               | 0.927566667 |    0.6875    | 0.626433333 |  0.57446667  |  0.52426667 |  0.46656667  |  0.40953333 |
> | FSGM STD                | 0.001650253 |  0.013865425 | 0.026476468 |  0.03488572  |  0.04250604 |   0.0445426  |  0.04344702 |
> | FSGM Targeted MEAN      | 0.927566667 |  0.719066667 |    0.6587   |    0.6042    |  0.54856667 |  0.48763333  |  0.42463333 |
> | FSGM Targeted STD       | 0.001650253 |  0.014600457 | 0.032930837 |  0.03766656  |  0.04145001 |  0.04438404  |  0.04252262 |
> | Iterative MEAN          | 0.927566667 |  0.618733333 | 0.500066667 |  0.42243333  |  0.36636667 |  0.32313333  |  0.29333333 |
> | Iterative STD           | 0.001650253 |  0.019620737 | 0.049166384 |  0.05432949  |  0.05828562 |  0.05640854  |  0.05413172 |
> | Iterative Targeted MEAN | 0.927566667 |  0.709766667 | 0.613233333 |    0.5412    |    0.4884   |  0.45046667  |  0.41603333 |
> | Iterative Targeted STD  | 0.001650253 |  0.020705152 | 0.045928459 |  0.05445062  |  0.06164033 |  0.05794259  |  0.05452654 |

---

> ### Author Response · Authors · 2020-11-13
> **Responses to Reveiwer 3 (Part 2)**
>
> - “What’s the performance with high dimensional and high entropy label representations, comparing to text label, for other kinds of the classification problems, such as NLP problems.”
>
> This is a great point. We absolutely agree that this could be the next application domain to study. Since our focus on this paper is on image classification tasks only, we leave the research on NLP tasks for future studies.
>
> - “For speech label, in model evaluation, what’s the performance for the model, if we choose a speech to text process to obtain its ground truth, besides the two approaches mentioned in the paper -- “nearest neighbor” and a validated loss threshold.”
>
> Our validation process of the threshold can also be seen as an instance of speech to text process evaluation. We detailed the evaluation process and results in Appendix A.1. Specifically, our evaluation process is a Forced-Choice Testing where each human subject needs to select the closest text label corresponding to the given speech predictions from our model. If the predicted speech is not clear, the human can select the additional “None” class. Overall, humans agree with the trained model 99.4% of the time when we count the workers independently, and 99.8% of the time when we take a majority vote. We did not use pre-trained speech to text labels to avoid the distribution shift introduced by different speech signal domains. Unlike natural speech signals commonly used on speech to text models, we chose a publicly available machine TTS system for controlled experiments to leave the exploration for different languages and intonations for future study. When the speech labels are also diverse natural signals, different speech to text models can also be alternative options.
>
> Please kindly let us know if we address your concerns. Thank you!

---

> ### Author Response · Authors · 2020-11-13
> **Responses to Reveiwer 3 (Part 1)**
>
> Thank you for your helpful suggestions and recognition of our novel perspective on underexplored problems. We appreciate your support and detailed understanding of our work. Please kindly find our point response below:
>
> - “The clarity of the study on the underline true set of characteristics that contribute to the improvement, from speech label, shuffled-speech label, Gaussian-composition label, besides high dimension and high entropy.” “It could be valuable to future investigate the inherent property that contributes to the improvement, besides high dimensionality and high entropy.”
>
> Thank you for the encouragement for future investigation. We definitely agree that future studies are needed to better understand the theory behind our findings. In this paper, we tend to provide extensive experiments to demonstrate empirical findings. We hope that this paper will inspire both future empirical and theoretical work on the role of label representation in the learning process.
>
> We cannot immediately tell the exact inherent property that contributes to the final improvement. We try to provide some intuitions and analysis for future investigations. From our visualizations in Figure 5 and Figure 8, high-dimensional high-entropy labels seem to give more separate clusters of image class features both the early stage of the learning and the end of the learning. This suggests that high-dimensional and high-entropy labels aid in learning more distinct representations of the classes. As to other possible inherent properties of the labels, we certainly agree that this is a valuable future direction for work. From our set of experiments, the common characteristics do seem to be dimensionality and entropy, however there could very well be subtleties. Additionally, it could be the case that not all high-dimensional, high-entropy labels are equally beneficial. We believe this is also an interesting direction for future work.
>
>
> - “The logic behind the adversary image generation, target vs non-target.”
>
> Both target and non-target attacks are among the most popular adversarial attacks which can change the input image slightly with large prediction errors on neural networks. The perturbations are often not visibly by humans but have a large impact on neural networks. We chose them also because they are one of the first proposed attacks that are intuitive, easy to implement, and fast to apply. There are many other attacks that would be interesting to try and we leave other algorithms for future exploration.
>
> - “And the evidence/thinking process behind the pre-selected threshold of 3.5”
>
> Humans are very good at recognizing the semantic meaning of speech signals. One way to evaluate the prediction results is to play the predicted speech spectrograms to humans after converting them back to the original waveforms. As detailed in Appendix A.1, we indeed performed this evaluation on diverse human subjects. To further provide quantitative evaluations of our results and future research benchmark, we need to decide a threshold. We initially came up with this threshold by only running the tests on a small subset of humans with multiple possible threshold values. We found 3.5 works the best. We then further conducted large-scale validations on many more human subjects to validate the threshold evaluation.

---

> ### Author Response · Authors · 2020-11-20
> **Responses to Reviewer 3 (Part 5)**
>
> Dear Reviewer 3,
>
> We are happy to include more results as we have finished the additional experiments left last time. All the new experiments on BERT and GloVe suggest that our conclusion about high-dimensional high-entropy label representation remains the same. Hope our new experiments help address your questions above. Thank you!
>
> | BERT           |     1%     |     2%     |     4%     |     8%     |     10%    |     20%    |
> |----------------|:----------:|:----------:|:----------:|:----------:|:----------:|:----------:|
> | ResNet32 MEAN  |   0.36952  |   0.50222  |   0.65856  |   0.76822  |   0.7873   |   0.84386  |
> | ResNet32 STD   | 0.03582432 | 0.01305946 |  0.0090088 | 0.00463094 | 0.00490958 | 0.00362028 |
> | ResNet110 MEAN |   0.34586  |   0.44366  |   0.61492  |   0.76088  |   0.78096  |   0.83934  |
> | ResNet110 STD  | 0.03373886 | 0.01979602 | 0.02300882 | 0.00431481 | 0.00638736 | 0.00386088 |
> | VGG19 MEAN     |   0.3913   |   0.5287   |   0.67006  |   0.77818  |   0.79324  |   0.8494   |
> | VGG19 STD      | 0.04019353 |  0.0112394 | 0.01532783 | 0.00109618 | 0.00379768 | 0.00206978 |

---

### Official Review · AnonReviewer4 · 2020-10-29
**TTS-generated audio can be used to train classification model instead of one-hot class label.**

**Rating:** 7
**Confidence:** 4

**Review:**

The paper proposed to use other high-dimensional representation instead of direct one-hot class labels in image classification problem. They used spectrogram of the pronunciations (TTS-based generated speech) of class labels as a high-dimensional representation. Then, they performed regression using the spectrogram as a label. The evaluation is conducted in nearest-neighbor way, measuring the distance between the image embedding feature and groundtruth high-dimensional representation (spectrogram) and decide the predicted class label. Then, they compared it with traditional classification model with cross-entropy loss. The results showed that the proposed approach are more robust in adversarial attack and feature effectiveness.

However, overall, I think one important related direction is somewhat missing which is zero-shot learning. In zero-shot learning, the primary goal is to make a prediction on unseen class labels. On the other hand, zero-shot learning has characteristics in that it uses an additional information to learn the embedding space. I think the proposed approach in this paper is somewhat related with it.
Since TTS system is utilized to generate pronunciation of label, such other information is naturally used in high-dimensional labels (the information used in the TTS system). Also, the author argues that audio labels is special, but in the paper, other type of high-dimensional representation of labels that uses external information are not explored, such as word2vec.

I think two factors are somewhat mixed. One is about the usefulness of high-dimensional representation (without external information, constant comparison is working at here) and the other is use of the external information. So, to verify the former factor, I think high-dimensional version of class label that does not use the external information should be added, such as after running topic modeling algorithm within the image classification data, and use them as a high-dimensional representation of a class label. Second, to verify the specialness of the audio label, other external data can also be compared, such as general word2vec, ...

In page 4, how the various length of audio produce the spectrogram of the same size? (the length difference is really small?)

The traditional classification is conducted in classification, while the high-dimensional label experiment is conducted in regression. I think the author can explore classification type loss for the latter experiments also (measuring the distance between the two matrices and put softmax over these similarity scores).

---

> ### Author Response · Authors · 2020-11-13
> **Responses to Reviewer4 (Part 4)**
>
> ##### BERT Attack Results:
>
> | ResNet32                |  Epsilon 0  | Epsilon 0.05 | Epsilon 0.1 | Epsilon 0.15 | Epsilon 0.2 | Epsilon 0.25 | Epsilon 0.3 |
> |-------------------------|:-----------:|:------------:|:-----------:|:------------:|:-----------:|:------------:|:-----------:|
> | FSGM MEAN               | 0.927733333 |  0.690766667 | 0.607433333 |  0.51736667  |  0.42313333 |  0.33553333  |  0.26746667 |
> | FSGM STD                | 0.001625833 |  0.015631165 |  0.0345558  |  0.04923437  |  0.05716374 |  0.05364516  |  0.04588489 |
> | FSGM Targeted MEAN      | 0.927733333 |  0.738366667 | 0.670866667 |    0.5776    |  0.46483333 |  0.36613333  |    0.2873   |
> | FSGM Targeted STD       | 0.001625833 |  0.012159907 | 0.025550408 |  0.04388633  |  0.05841955 |  0.05822837  |  0.0491692  |
> | Iterative MEAN          | 0.927733333 |  0.616066667 |    0.4877   |  0.40496667  |  0.34303333 |    0.2986    |  0.25896667 |
> | Iterative STD           | 0.001625833 |  0.029737574 | 0.048071925 |  0.05809392  |  0.06387256 |  0.06849299  |   0.066176  |
> | Iterative Targeted MEAN | 0.927733333 |  0.711233333 | 0.631633333 |  0.57116667  |  0.51126667 |  0.46513333  |    0.4242   |
> | Iterative Targeted STD  | 0.001625833 |  0.02125049  | 0.035287722 |  0.04435903  |  0.05630385 |   0.0603089  |  0.0656461  |
>
> | VGG19                   |  Epsilon 0  | Epsilon 0.05 | Epsilon 0.1 | Epsilon 0.15 | Epsilon 0.2 | Epsilon 0.25 | Epsilon 0.3 |
> |-------------------------|:-----------:|:------------:|:-----------:|:------------:|:-----------:|:------------:|:-----------:|
> | FSGM MEAN               | 0.936866667 |    0.6461    |    0.5969   |  0.57206667  |  0.54673333 |  0.50866667  |  0.45506667 |
> | FSGM STD                | 0.000896289 |  0.006657327 | 0.003534119 |  0.00591383  |  0.01069268 |  0.00884892  |  0.00123423 |
> | FSGM Targeted MEAN      | 0.936866667 |    0.7262    | 0.684433333 |  0.65903333  |  0.63113333 |    0.5858    |  0.52106667 |
> | FSGM Targeted STD       | 0.000896289 |  0.012247857 | 0.006572924 |  0.01130059  |  0.0169512  |  0.01455026  |  0.01020016 |
> | Iterative MEAN          | 0.936866667 |  0.564933333 | 0.462633333 |  0.40366667  |    0.3629   |  0.33546667  |    0.317    |
> | Iterative STD           | 0.000896289 |  0.00280238  | 0.016914589 |  0.02437875  |  0.02829629 |  0.02835918  |  0.0265684  |
> | Iterative Targeted MEAN | 0.936866667 |  0.734033333 |    0.6511   |  0.58886667  |  0.54796667 |  0.51586667  |  0.48706667 |
> | Iterative Targeted STD  | 0.000896289 |  0.011402777 | 0.010247439 |   0.0160076  |  0.01668422 |  0.01494468  |  0.01494869 |
>
> ##### Data Efficiency
>
> | GloVe          |      1%     |      2%     |      4%     |     8%     |     10%    |     20%    |
> |----------------|:-----------:|:-----------:|:-----------:|:----------:|:----------:|:----------:|
> | ResNet32 MEAN  |    0.3141   |   0.40458   |   0.50112   |   0.62104  |   0.66468  |   0.77284  |
> | ResNet32 STD   | 0.017050161 | 0.011675855 | 0.010050552 | 0.00893501 | 0.00603569 | 0.00620729 |
> | ResNet110 MEAN |   0.27674   |    0.3636   |   0.46344   |   0.59132  |   0.6368   |   0.77408  |
> | ResNet110 STD  | 0.025722877 | 0.015070103 |  0.02634104 |  0.0180268 | 0.00697567 | 0.00309283 |
> | VGG19 MEAN     |   0.37956   |   0.47456   |    0.5894   |   0.7458   |   0.7759   |   0.84446  |
> | VGG19 STD      | 0.005315111 | 0.011771763 | 0.009727692 | 0.00290586 | 0.00634161 | 0.00480816 |

---

> ### Author Response · Authors · 2020-11-13
> **Responses to Reviewer4 (Part 3)**
>
> ##### GloVe Attack Results:
>
> | ResNet32                |  Epsilon 0  | Epsilon 0.05 | Epsilon 0.1 | Epsilon 0.15 | Epsilon 0.2 | Epsilon 0.25 | Epsilon 0.3 |
> |-------------------------|:-----------:|:------------:|:-----------:|:------------:|:-----------:|:------------:|:-----------:|
> | FSGM MEAN               | 0.908366667 |    0.5895    |    0.5178   |  0.43976667  |  0.35593333 |  0.28626667  |  0.23183333 |
> | FSGM STD                | 0.000971253 |  0.011358257 | 0.017911728 |  0.02619338  |  0.03930755 |  0.04576869  |  0.04143336 |
> | FSGM Targeted MEAN      | 0.908366667 |  0.663833333 |    0.5728   |  0.47496667  |  0.37633333 |  0.29713333  |    0.2388   |
> | FSGM Targeted STD       | 0.000971253 |  0.004045162 | 0.016937237 |  0.03755001  |  0.0529005  |  0.05715263  |  0.05345849 |
> | Iterative MEAN          | 0.908366667 |  0.536733333 | 0.406266667 |    0.3248    |  0.26723333 |  0.22213333  |  0.19023333 |
> | Iterative STD           | 0.000971253 |  0.005478443 |  0.00546382 |   0.0051643  |  0.01362583 |  0.01503806  |  0.02038929 |
> | Iterative Targeted MEAN | 0.908366667 |  0.654166667 | 0.558166667 |    0.4847    |    0.4289   |    0.3854    |    0.3449   |
> | Iterative Targeted STD  | 0.000971253 |  0.005201282 | 0.011447416 |  0.01825514  |  0.02280066 |   0.0317572  |  0.03139379 |
>
> | VGG19                   |  Epsilon 0  | Epsilon 0.05 | Epsilon 0.1 | Epsilon 0.15 | Epsilon 0.2 | Epsilon 0.25 | Epsilon 0.3 |
> |-------------------------|:-----------:|:------------:|:-----------:|:------------:|:-----------:|:------------:|:-----------:|
> | FSGM MEAN               | 0.927566667 |    0.6875    | 0.626433333 |  0.57446667  |  0.52426667 |  0.46656667  |  0.40953333 |
> | FSGM STD                | 0.001650253 |  0.013865425 | 0.026476468 |  0.03488572  |  0.04250604 |   0.0445426  |  0.04344702 |
> | FSGM Targeted MEAN      | 0.927566667 |  0.719066667 |    0.6587   |    0.6042    |  0.54856667 |  0.48763333  |  0.42463333 |
> | FSGM Targeted STD       | 0.001650253 |  0.014600457 | 0.032930837 |  0.03766656  |  0.04145001 |  0.04438404  |  0.04252262 |
> | Iterative MEAN          | 0.927566667 |  0.618733333 | 0.500066667 |  0.42243333  |  0.36636667 |  0.32313333  |  0.29333333 |
> | Iterative STD           | 0.001650253 |  0.019620737 | 0.049166384 |  0.05432949  |  0.05828562 |  0.05640854  |  0.05413172 |
> | Iterative Targeted MEAN | 0.927566667 |  0.709766667 | 0.613233333 |    0.5412    |    0.4884   |  0.45046667  |  0.41603333 |
> | Iterative Targeted STD  | 0.001650253 |  0.020705152 | 0.045928459 |  0.05445062  |  0.06164033 |  0.05794259  |  0.05452654 |

---

> ### Author Response · Authors · 2020-11-13
> **Responses to Reviewer4 (Part 2)**
>
> - “Second, to verify the specialness of the audio label, other external data can also be compared, such as general word2vec, ...”
>
> Thank you for the great suggestion. We are happy to report the newest experiments with GloVe and BERT embeddings on CIFAR10,  as noted above. To summarize, we find the results from our experiments on GloVe and BERT to be consistent with our previous results.
>
> - “In page 4, how the various length of audio produces the spectrogram of the same size? (the length difference is really small?)”
>
> After obtaining the spectrograms, we treat the spectrograms as 2D images and resize them to the same size. Since all the labels are single words, the differences in length are very small. We verify that this is a valid operation in two ways. First, we can invert the label generation process to waveform format and the speech label can be recovered without the loss of quality. Second, we can invert the label generation process on the predicted speech labels from our network. As shown in Appendix A.1 as well as Section 4, the produced speech label can obtain 99.8% agreement among human subjects under our threshold. We therefore conclude that our processing of speech labels is valid. This method is also commonly used in audio-visual learning, as pointed out in our references.
>
> - “The traditional classification is conducted in classification, while the high-dimensional label experiment is conducted in regression. I think the author can explore classification type loss for the latter experiments also (measuring the distance between the two matrices and put softmax over these similarity scores).”
>
> In our baseline comparison, we compared against label-smoothing techniques which build on the traditional cross-entropy classification loss. As shown in Figure 4 and Section 5.3, label-smoothing does not provide further benefits when the training data is limited. We thank the reviewer for this suggestion of an alternate loss function. We agree that there are other interesting losses that can be used for training while supervising with high dimensional labels, including some variant of a soft KNN in the high-dimensional output space. We believe that this is an important direction for future work, but not directly relevant to the purpose of this paper. Here, we just supervise with the most obvious choice of loss function between the predicted and ground-truth high-dimensional labels by posing the loss as a regression. We leave the study of the impact of alternative loss functions as a future study.
>
> Please kindly let us know if we address your concerns. Thank you!
>
> ----------------------
> References
> [1] Hockett, C. D. (1960). The origin of speech. Scientific American, 203(3), 88–96. https://doi.org/10.1038/scientificamerican0960-88

---

> ### Author Response · Authors · 2020-11-13
> **Responses to Reviewer4 (Part 1)**
>
> Thank you for your constructive comments and suggestions. We will try our best to address your points below:
>
> - “TTS-generated audio can be used to train the classification model instead of a one-hot class label.”
>
> We would like to clarify the main contribution of our paper. Although our initial motivation was testing whether speech labels provide benefits over categorical labels in the classification task, we ultimately show that speech is just one instance of label representation that yields benefits (in the form of data efficiency and adversarial robustness) over traditional categorical labels. In Section 4 “Other Labels” we list several other label representations that provide similar benefits to audio labels. Our main finding is that the common characteristics among the label representations that perform well are high-dimensionality and high-entropy. The performance of these various label representations is jointly plotted in Figures 2, 3, and 4.
>
> -  “One important related direction is somewhat missing which is zero-shot learning.”
>
> As the reviewer pointed out, zero-shot learning aims to “make a prediction on unseen class labels”. We would like to clarify that this is not our goal in this paper. Our goal is to study the impact of various label representations on the traditional image classification task rather than generalizing the learned features to unseen classes.
>
> - “Zero-shot learning has characteristics in that it uses additional information to learn the embedding space” “Other types of high-dimensional representation of labels that use external information are not explored, such as word2vec.”
>
> Although the speech labels are generated from the TTS system, which does assume such a system exists, we do not believe that the speech signals contain additional information about a category’s semantics. This is because the conventional view in linguistics is that language is arbitrary - the sound of a word contains no information about the words meaning [1].
>
> Additionally, the speech label is only one of the many label representations in our paper. The speech label serves as the inspiration for our further experiments in the rest of the paper. For example, we verify our observations on other high-dimensional labels including the composition of Gaussian patterns, random-uniform matrix, and constant matrix. None of these labels assume any other form of external semantic information other than the given dataset itself. Therefore, we believe that our conclusion about high-dimensional high-entropy label representations still holds under the absence of external semantic information.
>
> We agree that experimenting with labels that do contain semantic information is an interesting direction. In this spirit, we are happy to report new experiments with two additional representations: BERT word embedding and GloVe embeddings. Since we hoped to report as soon as possible given the limited time, we only finished the experiments on the CIFAR10 dataset and will complete all the experiments as they run.
>
> Specifically, we find that GloVe embedding labels (50-dim vector) outperform categorical labels in both data efficiency and adversarial robustness. The performance of the GloVe labels are comparable to that of the low-dimensional label experiment (10-dim) (see Figures 3 and 4).
> We also find that BERT embeddings labels (48x48 matrix) outperform both categorical labels and GloVe/low-dimensional labels. As a high-dimensional and high-entropy label, BERT embeddings also achieve comparable performance with our other high-dimensional high-entropy labels. These results are consistent with the observations in our paper. Namely, that high dimensionality and high entropy both improve feature representations. GloVe labels are higher entropy than categorical labels, but lower dimensionality/entropy than BERT labels.
>
> In summary, we find consistent results across speech, GloVe, BERT, composition of Gaussian patterns, uniform-random, and constant labels. Therefore, we believe our conclusion remains valid.
>
> - “I think a high-dimensional version of class label that does not use the external information should be added, such as after running topic modeling algorithm within the image classification data, and use them as a high-dimensional representation of a class label.”
>
> Thank you for your suggestion on alternative high-dimensional labels. We agree that this is an interesting choice for label representation. However, since the label representations presented in this paper do not make use of any clustering or learning algorithm, we do not believe this is a fair comparison.

---

> ### Author Response · Authors · 2020-11-20
> **Responses to Reviewer4 (Part 5)**
>
> Dear Reviewer 4,
>
> We are happy to include more results as we have finished the additional experiments left last time. All the new experiments on BERT and GloVe suggest that our conclusion about high-dimensional high-entropy label representation remains the same. As the second round of rebuttal begins today, please kindly let us know if we have addressed your concerns. We would love to try our best to address your concerns, so please feel free to let us know. Thank you!
>
> | BERT           |     1%     |     2%     |     4%     |     8%     |     10%    |     20%    |
> |----------------|:----------:|:----------:|:----------:|:----------:|:----------:|:----------:|
> | ResNet32 MEAN  |   0.36952  |   0.50222  |   0.65856  |   0.76822  |   0.7873   |   0.84386  |
> | ResNet32 STD   | 0.03582432 | 0.01305946 |  0.0090088 | 0.00463094 | 0.00490958 | 0.00362028 |
> | ResNet110 MEAN |   0.34586  |   0.44366  |   0.61492  |   0.76088  |   0.78096  |   0.83934  |
> | ResNet110 STD  | 0.03373886 | 0.01979602 | 0.02300882 | 0.00431481 | 0.00638736 | 0.00386088 |
> | VGG19 MEAN     |   0.3913   |   0.5287   |   0.67006  |   0.77818  |   0.79324  |   0.8494   |
> | VGG19 STD      | 0.04019353 |  0.0112394 | 0.01532783 | 0.00109618 | 0.00379768 | 0.00206978 |

---

### Author Response · Authors · 2020-11-24
**Common summary for revision and rebuttal**

We thank all reviewers for their valuable feedback and comments. We appreciate that the reviewers recognized the novelty of our findings and the thoroughness of our experiments. We updated the paper with the suggested revisions. Throughout the rebuttal period, we also gave extensive responses with detailed explanations and additional experiments to each individual reviewer.

We appreciate the word embedding label representation experiments suggested by R1, R3, and R4, and our experiments suggest that our conclusion still holds. We updated the paper to included these new experiments.

---

### Decision · Program_Chairs · 2021-01-07
**Final Decision**

**Decision:**

Accept (Poster)

**Comment:**

This paper proposes to use high dimensional representation for labels to strengthen the adversarial robustness of deep neural networks. Experimental results demonstrate that the proposed method improve adversarial robustness. All reviewer agree that the authors propose an interesting idea and this direction deserves further exploration. On the other hand, the reviewers also raise a serious question: There is a lack of explanation of why high dimensional representation of labels improve adversarial robustness. Therefore, it is not clear if the proposed method can defend refined attacks tailored to such dimensional label representation. The authors are highly encouraged to conduct deeper analysis, especially on the robustness against finer attacks.